# Effect of the Use of Purified Grape Pomace as a Fining Agent on the Volatile Composition of Monastrell Wines

**DOI:** 10.3390/molecules24132423

**Published:** 2019-07-01

**Authors:** Rocio Gil-Muñoz, María Dolores Jiménez-Martínez, Ana Belén Bautista-Ortín, Encarna Gómez-Plaza

**Affiliations:** 1Instituto Murciano de Investigación y Desarrollo Agrario y Alimentario, Ctra. La Alberca s/n, 30150 Murcia, Spain; 2Food Science and Technology Department. Faculty of Veterinary Science, University of Murcia, Campus de Espinardo, 30100 Murcia, Spain

**Keywords:** purified grape pomace, fining agents, volatile composition, wine

## Abstract

(1) Background: The lack of viable alternatives for the industrial exploitation of grape pomace is one of the reasons why it is considered a serious environmental pollutant. However, as a byproduct, it could be used as a fining agent, since previous studies have shown that it is able to eliminate undesirable substances in wine. However, the little information available does not describe its effect on wine aroma. (2) Methods: Purified grape pomace extracts were used for fining a red wine and their effect on the volatile compounds of the wine was assessed, comparing the results with those obtained with different commercial fining agents. (3) Results: The results showed how purified grape pomace decreased the total volatile content of a wine to a similar extent as other fining products, such as yeast extracts or gelatin. Among the different families of volatile compounds analyzed, only total esters and terpenes differed from the levels recorded for a control wine, being slightly lower. No statistical differences were found for the rest of the volatile compounds (alcohols, carbonyl, lactones, and acids) compared with the levels measured in control wine. (4) Conclusions: The results suggest that purified grape pomace could be used as a non-allergenic wine fining agent.

## 1. Introduction

Grape pomace, resulting from pressing before or after fermentation processes, is generated in large amounts in many parts of the world and is the main solid organic waste from wineries [1,2,3,4]. Several studies have been carried out regarding the use of purified grape pomace (PGP) as fining agent. Among them, Bindon et al. [5] reported that pomace byproducts have a high affinity for proanthocyanidins, while Guerrero et al. [6] showed that grape and apple fibers, when used as fining agents, retain tannins but reduce anthocyanins, total phenolics, and wine color density. Jiménez-Martínez et al. [7] used grape cell wall material as fining agent and found that it significantly reduced the wine phenolic content, a reduction of 44% to 64% in tannin levels allowing wine astringency to be reduced. In the same study Jiménez-Martínez et al. [7] found that the use of purified grape pomaces as a fining alternative diminished the histamine content and, particularly, the levels of ochratoxin A.

Volatile compounds play a role in wine sensory characteristics, either directly or through their physical and chemical reactions with other analytes present in wine. However, fining treatments can produce sensorial modifications that affect taste and aroma in wines. The removal of polyphenols or proteins through fining treatments and precipitation has been shown to produce modifications of the flavor balance [8] since macromolecules in the wine may strongly bind small molecules such as aroma compounds [9]. Therefore, aroma compounds may decrease through the direct adsorption of aroma molecules to the fining agent or due to the interaction of aroma compounds with grape macromolecules, whose removal during fining may contribute to lower the aroma perception [10,11]. Also, it must be taken into account that the partition of volatile compounds between liquid and gas phases is governed by the volatility and solubility of aroma compounds, properties that will probably be affected by other wine components such as polysaccharides, proteins, and polyphenols [12]. Processes affecting polyphenol aggregation therefore could also lead to a significant loss of aroma.

The changes in the volatile composition of a wine can be positive or negative, depending on the compound involved, leading to a smoother and more equilibrate wine or, on the contrary, to a less aromatic wine. Such results are much dependent on how the volatile compounds are affected. Moio et al. [13] reported that fining with bentonite (80 g/hL), potassium caseinate (60 g/hL), silica gel (10 g/hL), gelatin (30 g/hL), or activated charcoal (20 g/hL) induced a loss of aroma compounds in Falanghina wine. Sims et al. [14] used triangle tests to compare red and white wines fined with PVPP, casein, or gelatin, and found that wine treated with gelatin was significantly different from an unfined control wine, the fining treatment reducing the phenolic content and changing the sensory characteristics, but no significant differences were found between the control wine and wine treated with PVPP or casein. Lisanti et al. [15] reported that PVPP significantly decreased the concentrations of 4-ethylguayacol and 4-ethylphenol in tainted wines but also those of esters. However, the sensory outcome of both treatments was a decrease in the intensity of phenolic off-odors, and a significant increase in the intensity of “red fruit” odor, even though esters decreased their concentration.

Fining can alter wine aroma profiles by binding not only free but also bound aromatic compounds [9,13,16]. The extent of the interaction between fining agents and bound aromatic compounds also depends on the above mentioned factors, such as the physicochemical characteristics of the agent, the chemical nature of the compound, and possible interactions between volatiles and other macromolecules previously linked by the fining agent [13,17,18,19].

The purpose of this research was to investigate how the use of purified pomace affected the aromatic fraction of the wines and also to compare its effect with that produced by other fining agents commonly used in oenology.

## 2. Results and Discussion

### 2.1. Effect of Fining Agents on Total Volatile Composition in Wine

The different families of volatile compounds found in the wine are shown in Figure 1, and include alcohols, carbonyl compounds, esters, acids, terpenes, and norisoprenoids. Differences in the total concentration of volatile compounds were found between the PGP-treated wines and control wines, although differences were also found with other fining agents such as the yeast autolysate and the gelatin. However, differences in the volatile compounds of the control wine and those in the wines made with the rest of the fining agents applied were hardly noticeable. Several researchers have studied the effect of different fining agents on the volatile composition of wines with different results. For example, Geneix et al. [20] showed the influence of yeast walls on the fixation of ethyl esters in a reconstituted model wine. Castillo-Sánchez et al. [21] used other fining agents such as PVPP and gelatin to fine a wine after aging eight months in barrels; the results of the sensory evaluation showed that the aromatic fraction was intensified and the fining agents favoring the conservation of the organoleptic fraction of the wine [11]. Granato et al. [8] reported that the treatment of wine with bentonite resulted in a statistically significant removal of various compounds, in particular ethyl esters and fatty acids.

With respect to the different chemical families, the sum of alcohols, main responsible of fermentative aroma, showed no differences with control wine when PGP was applied. For the rest of the fining agents applied, only the use of bentonite decreased the content of total alcohols. Armada et al. [22] using bentonite in Albariño wines found a reduction of the concentration of these compounds by 33%. Furthermore, Salazar et al. [23] reported a diminution in major alcohols when bentonite was added at the end of alcoholic fermentation.

The ester concentration in wine was affected by PGP and other fining agents applied such as the yeast autolysate, bentonite, and the plant protein. However, the rest of the fining agents used did not produce statistical differences with respect to the control wine. Granato et al. [8] reported that lentil and pea proteins caused a 40–60% depletion of ester compounds. Working with model solutions, Vincenzi et al. [11] informed that bentonite removed ethyl esters, an effect that was intensified if proteins were added to the solution, specifically in terms of the longest carbon chains (from ethyl octanoate to ethyl decanoate).

In regards to the other studied aromatic families (acids, carbonyls, terpenes, and norisoprenoids) that also contribute to the varietal and fermentative wine aroma, they did not change when PGP was used as fining agent, and only differences in the concentration of terpenes and isoprenoids with respect to the control wine were detected, although the same effect was found for all the rest of fining agents studied in this research. In contrast with these results, the use of bentonite decreased the concentration of carbonyl compounds and fatty acids. Armada et al. [21] reported that the use of bentonite as fining agent produced a diminution of carbonyls and terpenes, the major compounds responsible for varietal and prefermentative aroma, similarly to the findings of our experiment. On the other hand, Vincenzi et al. [11] showed that bentonite also removed the fatty acids present in a model wine solution.

### 2.2. Effect of Fining Agents on the Individual Volatile Compounds in Wine

The results for the individual volatile compounds are shown in Table 1, Table 2, Table 3 and Table 4.

#### 2.2.1. Alcohols

The results for individual alcohol compounds are shown in Table 1. The main alcohols detected in all wines studied were 2-methyl-propanol, 3-methyl-butanol, hexanol, and 2-phenyl-ethanol. These alcohols may contribute positively to a wine’s bouquet at low concentrations, while their concentration increases with extended aging; however, excessive levels (>300 mg/L) can add negative characteristics to a wine, i.e., pungent odors. The concentration of these alcohols did not change due to the use of the all fining agents, except for 3-methyl-1-butanol, which decreased with all the fining treatments except in the wine treated with casein. The odor of this compound has been described as being like nail varnish [24]. PGP, yeast autolysate, and plant protein led to the highest decreases in this compound.

Some other individual compounds were also affected, such as 3-methyl-thiopropanol, a compound whose concentration decreased with all the fining agents used. This can be an important observation because this compound tends to impart some nuances of cauliflower and raw potato to a wine [25]. By contrast, other researchers have claimed that its presence in wine may be considered beneficial because it accentuates the toasted and fruity notes, and also causes a reduction of pyrazine and eucalyptus notes [26].

Guaiacol and 4-ethyl-guaiacol are oak-derived volatile compounds, as is eugenol, although this last compound may also arise directly from the grapes. The concentration of these three compounds was reduced by the yeast derivative, which may affect the oaky wine aroma. However, the same was not found for PGP, since it decreased the concentration of 4-ethyl-guaiacol but did not affect guaiacol and increased the concentration of eugenol. The fact that PGP can induce such an increase could be due to fact that grape skin (the origin of the pomace used in the experiment) provided the largest part of the varietal volatile compounds to the wine [27]. Hence, during contact time (in this case, 21 days) some of these compounds might be transferred from the PGP to the wine, as in the case of eugenol and 2-ethyl-hexanol, two alcohols present in the grape skins. De Torres et al. [28] macerated freeze-dried Moscatel and Airén pomaces during the elaboration of a white wine and reported that when the lyophilized grape pomace was added during fermentation, some aromatic fractions were increased in the final wine, resulting in an accentuation of the fruity and fresh notes. In a study on the recovery of aromatic aglycones from grape winemaking by-products, Muñoz-González et al. [29] reported that they could act as a source of glycosidic aroma precursors, which, after hydrolysis, would release interesting odorant compounds with high aroma quality and low odor thresholds.

Other minor alcohols of the wine also increased in concentration when other fining agents were applied, with significant differences from the levels found for the control wine. For example, the concentration of 2,3-butanediol increased when egg albumin was applied, as did the concentration of cis-3-hexenol when egg albumin, caseinate, bentonite, and gelatin were used.

Bentonite was the fining agent which reduced the alcohol fraction the most, perhaps because it eliminated most of the proteins from wine, eliminating the aromatic components fixed to these proteins at the same time [9].

An alcohol in wine that was not affected by the fining treatments was 2-phenyl-ethanol. Granato et al. [8] also indicated that the fining process had marginal effects on the levels of this alcohol.

One positive effect found in some of the fining treatments was the reduction in the concentration of 4-ethyl-phenol, whose formation is due to *Brettanomyces* yeasts, imparting undesirable odors to wines that are associated with horse stables, leather, or horse sweat [30]. Milheiro et al. [31] reported that activated carbon was a very efficient fining agent for removing 4-ethylphenol from red wines, providing a 75% decrease in its headspace concentration. Lower reductions were found when using egg albumin, isinglass, carboxymethylcellulose, and chitosan, all of which might be considered for treating wine contaminated with 4-ethyl-phenol. However, no decrease in the concentration of 4-ethyl-phenol with the fining agents assayed could be detected.

#### 2.2.2. Esters

Esters are secondary volatile compounds mostly derived from alcoholic fermentation, although some of them form part of the varietal aroma. Short-chain aliphatic ethyl esters add fruity characteristics to wine, whereas higher, longer-chained ethyl esters may contribute soapy, oily, or waxy notes [32]. The main esters detected in the wines studied were diethyl-succinate, ethyl-hexanoate, ethyl-octanoate, ethyl-decanoate, ethyl-hexadecanoate, and 3-methyl-1-butanol acetate (Table 2). The concentration of the last three esters decreased due to the effect of PGP, and only 3-methyl-1-butanol octanoate increased compared with control wine levels.

As stated before, PGP reduced the content of total esters in wine, especially those of long-chain ethyl esters. Sanborn [33] reported that a protein, wheat gluten, reduced the concentration of the lower aliphatic esters (i.e., ethyl-butanoate or ethyl-hexanoate) but the higher aliphatic ethyl esters (ethyl-decanoate, ethyl-dodecanoate, and ethyl-hexadecanoate) were more affected by bentonite, which led to the lowest concentrations of these compounds in Gewürztraminer wines. Our results also showed that bentonite also affected the concentration of these long-chain acids. As stated above, the effect of bentonite aroma compounds may be due to the direct adsorption of aroma molecules onto the bentonite, or due to an interaction of aroma compounds with grape proteins, thus contributing to the loss of aroma when those proteins are removed by bentonite fining [10]. Vincenzi et al. [11] reported that the presence of wine proteins in a solution treated with bentonite led to a higher loss of esters with the longest carbon chains (from ethyl-octanoate to ethyl-decanoate). They also reported that, in the case of the ethyl esters, the longer the hydrocarbon chain, the greater the extent of their removal, ranging from a loss of 20% for ethyl-butanoate to 84% for ethyl-dodecanoate. This relation between hydrocarbon chain length and the rate of removal suggests the involvement of hydrophobicity phenomena in determining the interaction.

Other authors also found a decrease in diethyl-succinate when bentonite was used [10], which agrees with our results using both bentonite and the yeast autolysate as fining.

Most of the ethyl esters were retained by the yeast derivative product, confirming the findings of Geneix [19], who noticed a retention of ethyl esters by yeast walls, the extent of which increased with the macromolecule concentration.

Esters were less affected by the animal-origin proteins caseinate and egg albumin, although the gelatin promoted a decrease in some esters. By contrast, other authors reported that prefermentation fining with gelatin may cause an increase in some aroma compounds [34], the exact reason for which is not clear, although one hypothesis concerns the presence of extra phenylalanine in the juice as a consequence of gelatin fining.

#### 2.2.3. Terpenes and Norisoprenoids

The concentrations of terpenes and norisoprenoids, compounds responsible for the varietal aroma in wines, are shown in Table 3. Although some of these compounds do not surpass the olfactory threshold, they may contribute to improving the aroma perception of the fruity, citrus, and floral aromatic series [35].

All of these compounds suffered a decrease in their concentration when wines were treated with PGP, except for α-terpineol, whose concentration did not significantly differ from those of the control wine, and β-ionone, whose concentration increased in the PGP-treated wines. β-Ionone, with its notes of violet, was the main norisoprenoid component present in all wines studied and whose origin is the skins of the grape [36]. Several researchers have reported that in both red and white grape pomaces, a significant concentration of volatile compounds are retained [37,38] and which therefore could be released during the fining process.

When the remaining fining agents were used, some individual compounds also decreased; for example, (+)limonene decreased in all cases except when gelatin was applied, α-terpinol decreased with all the fining agents used except in the case of caseinate and bentonite, and nerolidol decreased in all situations except when caseinate was used. Citronellol and β-damascenone were less affected by the fining treatments. Armada and Falqué [15] reported that bentonite (60 g/hL) significantly reduced the total concentration of monoterpenes and C13-norisoprenoids, especially linalool, geraniol, β-pinene, and (+) limonene in Albariño wine. Moio et al. [12] found that Falanghina wines fined with bentonite (80 g/hL) showed significant losses of linalool and geraniol. Meanwhile, other authors found that the overall effect of bentonite on monoterpenes was low and non-significant with the exception of β-damascenone.

#### 2.2.4. Carbonyls and Lactones

Most of the compounds detected in this fraction were transferred from the oak into the wine during aging, including furfural, 5-methyl-furfural, and 2-ethyl-5-methyl furan. Furfuryl compounds, with their pleasant aroma, are formed from the degradation of hemicellulose and lignin during oak toasting and are of sensory importance [39]. Their threshold perception is usually higher than the concentration commonly present in the wines, although they help to increase the perception of lactones ceded by the wood [40]. For this reason, it is advisable that they should not be reduced in excess [41].

Another aldehyde detected, benzaldehyde, decreased with all the fining treatments. For example, it was not detectable after using egg albumin and gelatin to fine the wine. Granato et al. [8] also reported that fining decreased the concentration of benzaldehyde by 55% to 75%, with gelatin being most effective in its removal. They also stated that benzaldehyde is associated with bitter almond notes that are often considered an undesirable trait.

Three lactones were detected in all wines studied: γ-butirolactone, trans-β-methyl-γ-octalactone, and cis-β-methyl-γ-octalactone. γ-butyrolactone is an important aroma compound in wines, and in our case, no differences compared to the control wine were found in its concentration when PGP and the rest of the agents were used. This lactone is mainly formed during alcoholic fermentation by an internal esterification reaction between an acid function and an alcohol function in the same molecule, but it may also arise directly from the grapes, where it contributes to the varietal aroma [42].

The other two lactones detected (cis and trans-β-methyl-γ-octalactone) are related to aging in wood, during which time the wine is enriched with the two β-methyl-γ-octalactone isomers [43], whose origin seems to be mainly related to the thermal degradation of wood lipids with their characteristic odor of coconut and wood [44], although they are also associated with wines that have an intense aroma of vanilla [45]. Among the different fining treatments, only PGP and the yeast derivative decreased the concentration of β-methyl-γ-octalactone, in this case, the cis-form.

#### 2.2.5. Acids

Figure 1 showed that the total concentration of fatty acids was not affected by PGP or by the use of other fining agents. The only statistically significant difference found was related to the yeast autolysate, which slightly increased the acid concentration. The effect of the fining agentes on individual fatty acids is shown in Table 5.

Even though the total acids content did not differ from that measured in the control wine, the concentrations of hexanoic, octanoic, and decanoic acids were seen to slightly decrease when PGP was used. Some of the other fining agents increased some of these individual compounds, such as octanoic acid when caseinate and gelatin were applied or decanoic acid when caseinate was used. By contrast, yeast autolysate reduced the butanoic, hexanoic, and nonanoic acid concentrations.

Bentonite did not have an effect on the removal of octanoic and decanoic acids, where the electrostatic repulsion involving the negative charge of bentonite may have predominated over the hydrophobicity interactions. Contrary to that, it has already been suggested that bentonite can interact with fatty acids in grape juices [46].

A PPC analysis, a powerful visualization tool that enables groupings to be observed within data, was conducted to check whether the wines could be separated according to the variables studied and, if so, to identify which were mainly responsible for any grouping found (Figure 2). The first two principal components explained 58% of the variance, and representation of these two principal components showed that all the replications of control wines together with plant protein and casein had lower values in component 1 than PGP, yeast autolysate, bentonite, and gelatin, mainly due to lower values of alcohols, fatty acids, and esters. Along component 2, PGP can be differentiated from the other fining agents on the basis of terpenes, norisoprenoids, and lactones. The results of this analysis confirmed the previous observation that PGP retained part of the volatile aroma compounds to a greater extent than fining agents, such as plant protein or casein, but similar to other common fining agents, such as yeast autolysates or bentonite.

## 3. Material and Methods

### 3.1. Isolation and Purification of Pomace Grape

Purified pomace grape extract (PGP) was obtained after the devatting and pressing steps of a standard red wine vinification. The grape pomace was separated from seeds with a scalpel, and then treated with ethanol (70%) to eliminate proanthocyanidins and other phenolic compounds. The purification method was carried out for 24 h in an orbital shaker at 150 rpm in the dark, and repeated twice. After this step, the purified grape pomace was washed with Milli-Q water, lyophilized, and ground to a fine particle size with a grinder (Junior S, Moulinex, Mayenne, France).

### 3.2. Wine Fining

A Monastrell wine from the 2015 vintage aged in French oak barrels for three months was used in this experiment. Different fining agents were added to the wine. Treatments were conducted in triplicate. A control wine with no added fining agent was also prepared. Purified grape pomace (PGP) was used at a dose of 13 mg/mL and its effect was compared with seven different commercial fining agents, tested at the maximum dose recommended by the respective manufacturers. These fining agents included sodium bentonite (0.75 g/L, Enoproma, Murcia, Spain), potassium caseinate (0.75 g/L, Laffort SA, Bordeaux Cedex, France), an egg albumin powder (Ovovin, 0.15 g/L), a gelatin with low degree of hydrolysis (Vinigel Platinum, 0.10 g/L), a vegetal protein from pea (Proveget 100, 0.20 g/L), and a mannoprotein-rich yeast autolysate (SuperBouquet, 0.5 g/L). The last four products were obtained from Agrovin S.A. (Alcazar de San Juan, Spain). Aliquots (100 mL) of the wines were mixed with the fining agents and analyzed after 21 days.

### 3.3. Determination of Volatile Compounds in Wines after Fining

Volatile compounds (alcohols, esters, acids, phenols, carbonyls, terpenes, and norisoprenoids) were analyzed by solid-phase microextraction (SPME) and gas chromatography–mass spectrometry [47] using an HP 5890 CG computer coupled to a single quadrupole mass spectrometer HP 5972 (Agilent Technologies, Santa Clara, CA, USA). The SPME holder and fiber were from Supelco (Aldrich, Bornem, Belgium). The samples were prepared using 10 mL of wine, 3 g of NaCl, and 10 μL of 2-octanol standard (250 μg/L). The vial was tightly sealed with a PTFE-lined cap. The solution was homogenized with a vortex shaker (IKA, Konigswinter, Germany) and then loaded onto a Gerstel autosampling device (Gerstel GmbH & Co. KG, Mellinghofen, Germany). The autosampling program consisted of swirling the vial at 500 rpm, an equilibrium time of 15 min at 40 °C, thereafter inserting the fiber into the headspace for 30 min at 40 °C, and transferring the fiber to the injector for desorption at 240 °C for 5 min. An HP Innowax 30M capillary column (50 m × 0.32 mm, 0.25 μm thick, Agilent Technologies, CA, USA) was used for the analysis. Injections were done in the splitless mode for 0.75 min, using a 2 mm i.d. non-deactivated direct liner for the SPME (Agilent Technologies). The carrier gas was Helium 5.0 (Abelló Linde S.A., Barcelona, Spain) with a column head pressure of 8 psi. The oven temperature was programmed at 40 °C for 5 min, raised to 225 °C at 3 °C/min, and then held at that temperature for 5 min.

The MS was operated in electron ionization mode at 70 eV and in SCAN mode (mass range 50–200 amu) with the transfer line to the MS system being maintained at 240 °C. Peak tentative identification was carried out by comparing mass spectra with those of the mass library (Wiley 6.0, Chichester, UK) and comparing the calculated retention indices with those published in the literature. Semi-quantitative data were obtained by calculating the relative peak area (total ion count signal or that of selected fragments in the case of some co-eluted compounds) in relation to that of the internal standard. The peaks were identified by comparison of the mass spectrum with a library (Wiley 6.0) and comparing the calculated retention indices with those calculated under the same chromatographic conditions and with those mentioned in the bibliography. The semiquantitative data were obtained by calculating the relative area of the peak in relation to the internal standard. The samples were analyzed in triplicate and the concentration of each of the volatile compounds was taken as the average value.

### 3.4. Statistical Analysis Treatment

Significant differences were assessed by analysis of variance (ANOVA). This analysis, together with the principal component analysis (PCA), was performed using Statgraphics 5.0 Plus software (Statistical Graphics Corp., Rockville, MD, USA). A Tukey test was used to separate the means (*p* < 0.05) when the ANOVA tests were significant.

## 4. Conclusions

Fining agents are necessary in order to remove undesirable substances from wine, although they may affect the final quality of wines by altering the volatile compound concentrations.

In an attempt to add value to winery byproducts, PGP has been proposed as a fining agent due to its ability to adsorb some undesirable wine compounds. However, its behavior in regards to wine volatile compounds had not been fully studied. The results obtained in the present study showed how PGP decreased the wine total volatile compound concentration to a similar extent as other commercial products, such as yeast autolysate and gelatin. When the different families were studied, only the sum of esters and terpenes showed a significant reduction after PGP treatment.

On the other hand, PGP tended to increase the amount of some aromatic compounds, such as eugenol, probably due to the presence of these compounds in the purified pomace and their subsequent release. The results, therefore, pointed to the idea that this product could be used as a non-allergenic wine fining agent due to its ability to decrease the levels of undesirable compounds in wine, together with its limited activity when it comes to removing aromatic compounds. However, its use would need to be optimized since a loss of significant amount of wine volume through lees occurs following PGP addition. An option that will be tested is the repassing of wine over a fix bed of PGP to avoid wine losses.

## Figures and Tables

**Figure 1 molecules-24-02423-f001:**
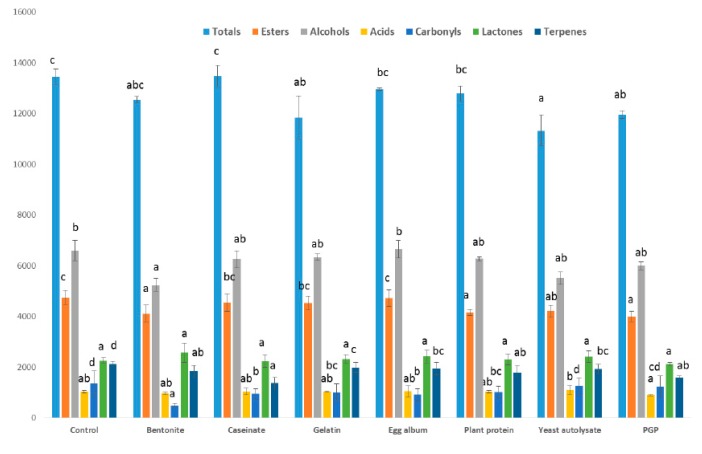
Effect of the fining agents and the purified grape pomace on the different families of aromatic compounds in wine (expressed as μg/L).

**Figure 2 molecules-24-02423-f002:**
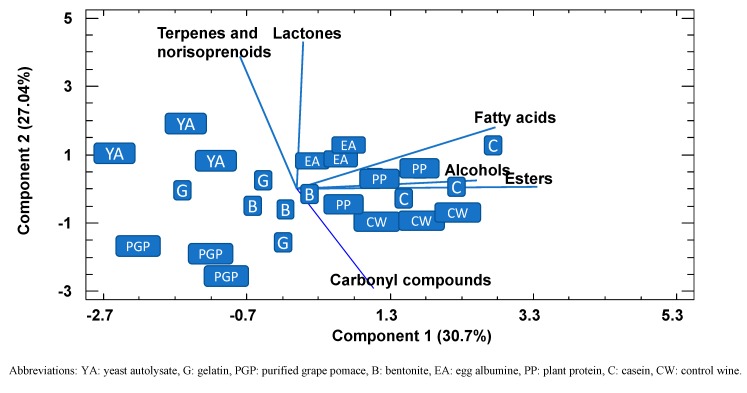
Bidimensional plot of the different fined wines using the first two components resulting from a principal component analysis using the different families of the volatiles compounds as variables.

**Table 1 molecules-24-02423-t001:** Effect of the fining agents on fusel alcohol compounds (µg equivalents of 2-octanol/L).

	RI	Control	PGP	Yeast Autolysate	Plant Protein	Egg Albumen	Caseinate	Bentonite	Gelatin
2-Methyl-propanol	1074	154.60 a	193.67 a	181.33 a	192.47 a	200.00 a	202.19 a	207.29 a	158.00 a
Butanol	1117	11.50 b	10.40 b	11.40 b	6.98 ab	11.26 b	10.09 b	3.38 a	10.00 b
3-Methyl-butanol	1184	3599.57 e	3139.07 ab	3.067.63 a	3275.66 abc	3336.55 bc	3540.83 de	3323.87 bcd	3348.80 cd
4-Methy-pentanol	1281	2.00 a	2.87 a	1.95 a	1.36 a	2.25 a	1.84 a	2.20 a	1.28 a
3-Methyl-1-pentanol	1294	17.80 a	21.35 a	21.25 a	22.86 a	21.87 a	17.85 a	23.80 a	15.66 a
Hexanol	1318	169.64 a	147.37 a	146.38 a	153.72 a	155.85 a	160.93 a	159.68 a	158.70 a
cis-3-Hexenol	1340	1.66 b	3.14 c	2.69 bc	2.81 bc	3.39 c	3.05 c	3.42 c	0.12 a
2-Ethyl-hexanol	1448	15.88 abc	24.09 d	12.95 ab	14.17 abc	16.47 bc	16.78 abc	18.94 cd	11.15 a
2,3-Butanediol	1488	37.22 a	35.45 a	31.26 a	33.25 a	35.85 a	40.07 a	36.43 a	28.50 a
4-Methyl-guaiacol	1573	28.47 bc	12.33 a	17.89 ab	16.11 a	17.06 ab	20.98 abc	21.78 abc	30.75 c
3-Methyl-thiopropanol	1650	21.09 d	15.33 b	12.56 a	15.41 b	15.01 b	18.12 c	18.59 c	17.49 c
Guayacol	1786	10.92 b	10.40 b	7.20 a	9.52 ab	10.35 b	10.79 b	9.67 ab	10.31 b
2-Phenyl-etanol	1837	2369.95 a	2243.81 a	1459.02 a	2344.40 a	2356.94 a	2460.00 a	2293.40 a.	1622.73 a
4-Ethyl-guaiacol	1949	24.39 b	19.95 ab	13.67 a	21.36 b	21.50 b	24.03 b	22.08 b	23.54 b
4-Propyl-guaiacol	2027	7.44 a	6.30 a	7.92 a	7.77 a	9.04 a	9.00 a	10.30 a	9.46 a
Eugenol	2083	14.00 b	21.67 c	9.90 a	13.29 ab	14.20 b	14.65 b	13.31 ab	15.56 b
4-Ethyl-phenol	2095	88.80 ab	73.35 ab	59.59 a	85.66 ab	89.23 ab	88.82 ab	74.53 ab	99.66 b

PGP: Purified grape pomace. Different letters within the same row and for each treated wine indicate significant differences, according to an LSD test (*P* < 0.05). RI: Retention index.

**Table 2 molecules-24-02423-t002:** Effect of the fining agents on ester compounds (µg equivalents of 2-octanol/L).

	RI	Control	PGP	Yeast Autolysate	Plant Protein	Egg Albumen	Caseinate	Bentonite	Gelatin
Ethyl-butanoate	1003	63.50 b	45.48 ab	42.69 ab	54.06 ab	52.72 ab	46.82 ab	53.81 ab	36.61 a
Ethyl-2-methyl-butanoate	1017	11.25 b	8.11 b	8.35 b	11.29 b	7.73 b	7.56 a	8.99 b	4.72 a
Ethyl-3-methyl-butanoate	1033	11.72 a	10.93 a	10.30 a	12.68 b	11.25 a	9.88 a	11.47 a	7.96 a
3-Methyl-butyl-acetate	1094	295.14 b	168.75 a	214.37 ab	258.98 ab	238.23 ab	235.89 ab	269.24 ab	171.76 a
2-Methyl-butyl-acetate	1109	4.21 b	2.59 a	3.01 ab	8.03 b	3.78 ab	1.66 a	10.29 b	2.67 a
Ethyl-hexanoate	1201	466.87 b	378.33 ab	377.00 ab	418.86 ab	434.00 b	417.24 ab	461.74 b	179.87 a
Ethyl-lactate	1303	242.05 b	211.21 ab	205.60 ab	216.25 ab	222.98 ab	252.18 b	214.91 ab	145.99 a
Ethyl-octanoate	1400	679.28 c	540.36 a	578.63 ab	674.58 c	645.97 bc	641.29 bc	607.41 abc	667.95 c
Ethyl-nonanoate	1494	55.15 abc	54.65 abc	49.29 a	55.33 abc	59.01 bc	60.73 c	52.58 ab	55.66 abc
3-Methyl-butyl-methoxy-acetate	1520	32.12 abc	34.71 bc	33.24 abc	31.08 ab	33.72 c	29.78 a	35.29 bc	34.66 bc
Ethyl-furan-2- carboxylate	1561	3.27 b	4.97 bc	0.0 a	0.0 a	5.12 c	3.33 b	4.96 bc	4.79 bc
Ethyl-decanoate	1598	433.79 d	160.64 a	302.23 b	371.69 c	344.42 c	438.01 d	379.00 c	374.29 c
3-Methyl-butyl-octanoate	1616	5.61 a	21.39 cd	24.87 de	19.47 c	25.78 de	11.69 b	8.00 ab	27.61 e
Diethyl-succinate	1623	2090.34 cde	2025.51 bc	1961.99 b	2052.23 cd	2068.65 cde	2144.80 e	1697.84 a	2131.45 de
Methyl-salicilate	1697	20.07 abc	14.84 a	15.51 ab	19.50 abc	21.60 abc	34.64 d	22.85 bc	24..05 c
Ethyl-benzyl-acetate	1719	13.78 ab	12.58 a	13.69 ab	13.73 ab	13.80 ab	14.40 ab	12.73 a	15.30 b
2-Phenyl-ethyl-acetate	1747	91.01 bc	70.10 ab	84.41 a	90.86 bc	93.80 c	99.15 c	89.68 bc	97.87 bc
Ethyl-hexadecanoate	2360	146.47 d	115.01 c	58.82 a	89.23 bc	73.52 ab	113.88 c	85.57 bc	67.83 a
Ethyl-hydrogen-succinate	2440	131.12 c	89.56 b	81.99 ab	61.14 a	62.95 a	82.81 ab	115.84 c	71.53 ab

PGP: Purified grape pomace. Different letters within the same row and for each treated wine indicate significant differences, according to an LSD test (*P* < 0.05). RI: Retention index.

**Table 3 molecules-24-02423-t003:** Effect of the fining agents on terpenes and isoprenoid compounds (µg equivalents of 2-octanol/L).

	RI	Control	PGP	Yeast Autolysate	Plant Protein	Egg Albumen	Caseinate	Bentonite	Gelatin
(+) Limonene	1345	3.40 c	0.88 a	2.15 b	0.94 a	0.34 a	0.45 a	1.17 ab	3.30 c
β Ionone	1470	76.97 b	409.97 cd	100.33 bcd	48.12 a	117.00 b	82.83 b	82.76 b	90.66 bc
Linalol	1504	24.21 bc	19.06 a	25.97 c	23.05 ab	23.10 ab	22.37 ab	22.99 ab	23.02 ab
α-Terpineol	1595	21.59 f	15.13 def	3.76 a	9.24 abc	6.50 ab	20.83 ef	17.83 ef	12.98 ced
β-Citronellol	1714	16.72 cd	10.44 a	14.26 b	15.95 bc	16.65 cd	16.67 cd	15.98 bc	18.09 d
β-Damascenone	1754	15.73 bc	10.19 a	18.91 c	13.98 abc	18.77 c	12.80 ab	15.66 bc	15.43 bc
Nerolidol	2082	40.10 e	31.47d	2.10 a	27.31 cd	17.66 b	39.89 e	24.00 c	16.83 b

PGP: Purified grape pomace. Different letters within the same row and for each treated wine indicate significant differences, according to an LSD test (*P* < 0.05). RI: Retention index.

**Table 4 molecules-24-02423-t004:** Effect of the fining agents on phenol and carbonyl + lactone compounds (µg equivalents of 2-octanol/L).

	RI	Control	PGP	Yeast Autolysate	Plant Protein	Egg Albumen	Caseinate	Bentonite	Gelatin
Furfural	1404	13.91 b	10.55 ab	10.05 a	10.24 a	13.14 b	12.41 ab	12.50 ab	11.49 ab
2-Ethyl-5-methyl furan	1444	4.33 abc	7.22 c	3.54 ab	3.10 ab	8.10 c	5.55 abc	6.98 bc	2.20 a
Benzaldehyde	1456	15.76 e	4.13 b	7.33 c	8.50 cd	0.00 a	1.93 ab	10.61 d	0.00 a
5-Methylfurfural	1509	102.37 b	102.12 b	75.97 a	75.05 a	80.76 a	72.48 a	71.98 a	113.64 b
γ-Butirolactone	1552	17.77 a	17.91 a	18.98 a	18.85 a	21.48 a	15.72 a	19.19 a	18.48 a
trans-β-Methyl γ octalactone	1817	83.17 a	82.13 a	76.83 a	83.94 a	80.53 a	84.94 a	87.80 a	77.60 a
cis- β-Methyl γ octalactone	1881	126.49 c	113.79 a	115.46 ab	122.11 bc	126.61 c	133.92 d	124.61 c	87.86 d

PGP: Purified grape pomace. Different letters within the same row and for each treated wine indicate significant differences, according to an LSD test (*P* < 0.05). RI: Retention index.

**Table 5 molecules-24-02423-t005:** Effect of the fining agents on acid compounds (µg equivalents of 2-octanol/L).

	RI	Control	PGP	Yeast Autolysate	Plant Protein	Egg Albumen	Caseinate	Bentonite	Gelatin
Acetic acid	1413	232.09 a	279.43 a	242.77 a	247.54 a	245.17 a	179.85 a	206.89 a	218.40 a
Butanoic acid	1586	18.75 b	18.14 ab	16.12 a	17.11 ab	17.45 ab	16.92 ab	16.55 b	17.37 ab
Hexanoic acid	1794	252.48 bc	213.95 a	215.58 a	229.83 ab	229.50 abc	268.99 c	254.47 bc	246.00 abc
Octanoic acid	1997	466.67 bc	353.50 a	437.24 b	474.20 cd	474.68 cde	507.55 de	485.67 cde	529.40 e
Nonanoic acid	2212	15.07 bc	10.66 abc	8.25 a	12.93 abc	17.96 c	14.76 bc	15.20 bc	19.71 c
Decanoic acid	2312	50.95 b	19.80 a	54.72 b	51.12 b	57.39 b	62.44 c	56.94 b	65.45 c

PGP: Purified grape pomace. Different letters within the same row and for each treated wine indicate significant differences, according to an LSD test (*P* < 0.05). RI: Retention index.

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
