# Peer review of "Effect of the Use of Purified Grape Pomace as a Fining Agent on the Volatile Composition of Monastrell Wines"

_molecules, 2019, doi:10.3390/molecules24132423_

Round 1
Reviewer 1 Report
General comments.
Please check for spacing and punctuation throughout the manuscript.
Manuscript needs language care.
Title
I would like to see the words “and fining agents” in the title. The manuscript describes comparison between different fining agents and grape pomace, yet the words “fining agents” are not in the title. It is misleading.
Organisations
Question” How does “Faculty of Veterinary Science” link up with Oenology?
Introduction
Rather use “concentrations” instead of “amounts” (Line 31 for example).
Rather use “affect/effect” instead of “influence” (Line 51 for example).
Avoid words like “high”, “much”, “they”, “also”, “very”, “interesting”, “our”, “then”, “its”
Line 46: …compounds cannot be “lost”
Line 48: …the loss of aroma… (Same comment as for line 46.
Line 55: What is an “elegant” wine? Line 56: What is an “attractive” wine?
Line 60: Make sure you use the word “observe” correctly. It is better to use the “found” when you actually have quantified something. Check the entire document.
Line 64: The word “contaminated” is not a suitable word.
Results and Discussion (check the spelling of discussion)
Line 90: Was the removal of compounds statistically significant?
Line 102: Promoted is not a suitable/correct word to use. Rather use the word “cause” A 40-60% decrease/depletion (not loss).
Line 104: Use the word “specifically” rather than the word “especially”
Lines 106-110: References are needed here. Which other studies?
Line111: “In contrast with these results”. Which results? If it refers to the paragraph of lines 106-110, it should not be two different paragraphs.
Line 117: Rather use the word “compounds”
Tables 1, 2, 3, 4, 5
Usually ANOVA test the differences between treatments. You mention different letters in the same column indicate significant differences according to ANOVA. The columns represent the measured variables whereas the rows represent the different treatments. Is it correct? There should be differences between measured variables because it is different compounds.
Line 161: What do you mean by….would be dragged?
Line 162: What is an important alcohol? Delete the word “important”
Line 167: …horse stables….
Lines 232-235: Rewrite this section. How high is quite high? “usually higher?...they help to increase? Not scientific terminology!
Line 236: ….“was seen to decrease…. Rewrite please.
Line 239: …is associated with bitter…...
Line 240:…undesirable….
Line 265: Bentonite treatment cannot completely remove decanoic and dodecanoic acids.
Line 265: dodecanoic acid is not listed in table 5.
Materials methods
Line 281: 3.1 Isolation and purification of pomace grape
Line 314: …Helium…
Line 309: Rather use the word “thereafter” or “where after” instead of “then”
Line 321: ….data were…..
Line 326: …..data was……
Line 329: 3.4 Statistical analysis
Conclusions
Line 345: What does …lend weight” mean?
References
There are many typing errors in the list of references. The hyphens between page numbers are different. Spacing needs to be checked.
Line 360: Should the name of the journal be abbreviated?
Line 369: The year (2018) occurs twice.
Lines 374-375: The title of this paper is not written as the others. All letters should be written in the lower case except the first letter of the first word.
Lines 379-380: The title of this paper is not written as the others. All letters should be written in the lower case except the first letter of the first word.
Lines 382-383: Vitis is spelt incorrectly. The journal abbreviation is incorrect. See line 396.
Line 398: Saccharomyces cerevisiae should be written in italics.
Line 406: 2017 should be in bold.
Line 432: Chardonnay not chardonnay, Gewurztraminer with a capital “G”, journal abbreviation incorrect. See line 396.
Line 452: Journal name should be in italics.
Lines 453-454: The first letter of every word of the title of the textbook should in capital letters.
Line 459: Quercus robur should be in italics.
Line 463: Line 452: Journal name should be in italics.
Author Response
ANSWERS TO REFEREE 1
General comments.
1. Please check for spacing and punctuation throughout the manuscript.
The spacing and punctuation have been checked throughout the whole manuscript, so we think that everything is right now.
2. Manuscript needs language care.
The language of the whole manuscript has been revised by native English, so we think that everything is right now.
3. I would like to see the words “and fining agents” in the title. The manuscript describes comparison between different fining agents and grape pomace, yet the words “fining agents” are not in the title. It is misleading.
The title has been changed by “Effect of the use of purified grape pomace as a fining agent on the volatile composition of Monastrell wines” as the revisor suggested.
4. Organisations: Question” How does “Faculty of Veterinary Science” link up with Oenology?
Because the Food Deparment forms part of this Faculty.
5. Rather use “concentrations” instead of “amounts” (Line 31 for example). It has been changed in the paper in those cases that we have considered that the revisor was right
6. Rather use “affect/effect” instead of “influence” (Line 51 for example). It has been changed in the paper
7. Avoid words like “high”, “much”, “they”, “also”, “very”, “interesting”, “our”, “then”, “its”
The whole manuscript has been revised and the words named by the reviser have been substituted in some cases where it was possible.
8. Line 46: …compounds cannot be “lost”. It has been substituted by “decrease” in the paper.
9. Line 48: …the loss of aroma… (Same comment as for line 46). It has been substituted by “decrease” in the paper.
10. Line 55: What is an “elegant” wine? That word has been substituted by “equilibrate”
11. Line 56: What is an “attractive” wine? That word has been substituted by “aromatic”
12. Line 60: Make sure you use the word “observe” correctly. It is better to use the “found” when you actually have quantified something. Check the entire document.
The authors have checked the entire document, and the word “observe” has been substituted by “found”, “reported”, “informed” and “noticed”.
13. Line 64: The word “contaminated” is not a suitable word. That word has been substituted by “tainted”
14. Results and Discussion (check the spelling of discussion)
The spelling of discussion has been checked throughout the whole manuscript as the reviser has suggested.
15. Line 90: Was the removal of compounds statistically significant?
Yes, so the authors have included the word “statistically” in the sentence.
16. Line 102: Promoted is not a suitable/correct word to use. Rather use the word “cause” A 40-60% decrease/depletion (not loss).
As reviser has indicated, the word “promoted” has been substituted by “cause” and the word “loss” by “depletion” in the text.
17. Line 104: Use the word “specifically” rather than the word “especially”
The authors have use “specifically” rather than “especially” in the paper.
18. Lines 106-110: References are needed here. Which other studies? The authors think that references are not needed in this part of the text because “the other studies” corresponded to “our studies” showed in this paper.
19. Line111: “In contrast with these results”. Which results? If it refers to the paragraph of lines 106-110, it should not be two different paragraphs.
The results are referred to the previous paragraph, so, as the reviser suggests is not necessary two different paragraphs. Therefore, both paragraphs have been joined.
20. Line 117: Rather use the word “compounds”
The authors have substituted the word “composition” by “compounds”.
21. Tables 1, 2, 3, 4, 5. Usually ANOVA test the differences between treatments. You mention different letters in the same column indicate significant differences according to ANOVA. The columns represent the measured variables whereas the rows represent the different treatments. Is it correct? There should be differences between measured variables because it is different compounds.
It is not correct in tables, so the word “column” must be changed by “row”. Due to the tables are images inserted by the Editorial throughout the manuscript, it should follow these instructions.
22. Line 161: What do you mean by….would be dragged? This sentence has been rewritten in the text in order to be better understood.
23. Line 162: What is an important alcohol? Delete the word “important”. The authors have deleted the word “important” in the text.
24. Line 167: …horse stables….The authors have added the word “horse” before “stables” in the text.
25. Lines 232-235: Rewrite this section. How high is quite high? “usually higher?...they help to increase? Not scientific terminology!
This sentence has been rewritten in the following way: “Their threshold perception is quite high - usually higher than the concentration commonly present in the wines”.
26. Line 236: ….“was seen to decrease…. Rewrite please. “Was seen to decreased” has been substituted by “decreased”
27. Line 239: …is associated with bitter…...The authors have substituted the word “related” by “associated” in the text.
28. Line 240:…undesirable….…...The authors have substituted the word “non-desirable” by “undesirable” in the text.
29. Line 265: Bentonite treatment cannot completely remove decanoic and dodecanoic acids.
This sentence has been rewritten in the following way in the text: “Bentonite did not have an effect on the removal of octanoic and decanoid acids, where the electrostatic repulsion involving the negative charge of bentonite may have predominated over the hydrophobicity interactions”
30. Line 265: dodecanoic acid is not listed in table 5. “Dodecanoic acid” has been eliminated in the sentence.
31. Line 281: 3.1 Isolation and purification of pomace grape. The authors have eliminated the words “of purified” in this sentence
32. Line 314: …Helium…Helium has been written in upper case.
33. Line 309: Rather use the word “thereafter” or “where after” instead of “then”. The authors have substituted the word “then” by “thereafter” in this line.
34. Line 321: ….data were…..It has been left with “were”.
35. Line 326: …..data was……The verb “was” has been substituted by “were”.
36. Line 329: 3.4 Statistical analysis. The authors have substituted the word “data” by “analysis” in this line.
37. Line 345: What does …lend weight” mean? The authors have substituted the words “lend weight” by “pointed” in the text.
38. There are many typing errors in the list of references. The hyphens between page numbers are different. Spacing needs to be checked.
The hyphens between page numbers have revised and changed in those cases where there was a mistake.
39. Line 360: Should the name of the journal be abbreviated? It has been included the name of the journal in abbreviated way.
40. Line 369: The year (2018) occurs twice. The year has been eliminated once from the paper.
41. Lines 374-375: The title of this paper is not written as the others. All letters should be written in the lower case except the first letter of the first word.
The title of this paper has been written as the other yet.
42. Lines 379-380: The title of this paper is not written as the others. All letters should be written in the lower case except the first letter of the first word.
The title of this paper has been written as the other yet.
43. Lines 382-383: Vitis is spelt incorrectly. The journal abbreviation is incorrect. See line 396.
Vitis has been spelt correctly and the journal abbreviation has been corrected as well.
44. Line 398: Saccharomyces cerevisiae should be written in italics. Saccharomyces cerevisiae has been written in italic.
45. Line 406: 2017 should be in bold. 2017 has been written in bold.
46. Line 432: Chardonnay not chardonnay, Gewurztraminer with a capital “G”, journal abbreviation incorrect. See line 396. Chardonnay and Gewurztraminer has been written with capital letter, and the abbreviation of the journal has been corrected.
47. Line 452: Journal name should be in italics. Journal name has been rewritten in italics.
48. Lines 453-454: The first letter of every word of the title of the textbook should in capital letters. The first letter of the words of the tittle has been rewritten in capital letters.
49. Line 459: Quercus robur should be in italics. Quercus robur has been rewritten in italics.
50. Line 463: Line 452: Journal name should be in italics. Journal name has been rewritten in italics.
Reviewer 2 Report
The manuscript entitled, "Effect of purified grape pomace on the volatile composition of Monastrell wines", follows in the series of papers from the authors looking into the effects of grape pomace as a fining agent and its efficacy compared to commercial fining agents. The manuscript presents the impacts on the volatile aromas of the various fining agents. The research is well presented and will be of interest to industry and academics alike.
The manuscript presents the data in a manner that is comprehensive. The authors support their results with relevant and thorough discussion. The format of addressing the impacts on the aroma groups (esters, alcohols, etc.), rather than all aromas as a whole, provides the material in a manner for quick interpretation. This presentation style also allows the authors to isolate key aromas and discuss the impacts in greater detail without confounding the discussion with other non-group compounds.
The authors' approach to presenting their finding does lack a presentation of the impact across all the groupings. They bring their findings together for a single paragraph looking at the differences between fining treatments on the entire volatile aroma landscape of the wines. A bit more discussion on the total impact on aromas would have been appreciated.
Overall the manuscript needs only minor corrections. Firstly, the figures and tables are quite small when printed. Tables placed into landscape format and figures utilizing the entire page width would allow for ease of reading. Second, the manuscript requires minor English editing. The manuscript is therefore recommended for publication
Author Response
1. The authors' approach to presenting their finding does lack a presentation of the impact across all the groupings. They bring their findings together for a single paragraph looking at the differences between fining treatments on the entire volatile aroma landscape of the wines. A bit more discussion on the total impact on aromas would have been appreciated.
The authors think that the impact of the different groups of aromas is well explained throughout the manuscript, so perhaps it is not necessary to deep more in this first part of the paper.
2. Overall the manuscript needs only minor corrections. Firstly, the figures and tables are quite small when printed. Tables placed into landscape format and figures utilizing the entire page width would allow for ease of reading. Second, the manuscript requires minor English editing. The manuscript is therefore recommended for publication.
The editorial has included the figures and tables in this format, so it must change it if it considers possible for the publication of this paper.
Reviewer 3 Report
The evaluation of the effect on aroma profile of the new fining agent PGP is a very interesting topic. In fact the exploitation of new non-allergenic sources for wine fining is of great interest.
The paper is clear and well written, with just some typing error (highlighted below).
Just one general consideration let me uncertain about the possible commercial use of the PGP as fining agent, that is the very high dosage needed to have an effect. I’m sure this issue has been already faced in the previous papers published on this topic, however for honesty this aspect should be at least mentioned also in the present paper. For example, using 13 mg/mL what is the loss of wine volume due to the adsorption by the fining agent? This aspect should be mentioned in the conclusions, as a possible limit for the applicability of this new product.
Some additional suggestions:
Line 16: the sentence “ to further our knowledge in this respect” could be deleted
Line 30: substitute and/or with “before or after”
Line 33: are you sure of the correctness of the citation number 5?
Line 37: maybe is better “a reduction” instead of “the reduction”
Line 49 and 89: add the reference 23
Line 99, 117, 290, 335 finning à fining
Line 161: we observed
Line 179: concentration of the last three…
Line 190 (also in line 462): Vincenzi
Line 212: the
Line 247: delete the last sentence (is a repetition)
Line 265-268: these data confirm those of Vincenzi who found an interaction of bentonite with fatty acids that increase with the length of the chain
Line 273: all the replicates of control wine together with those treated with…
Line 274: maybe higher values (if it is the case, substitute also in line 275). What about the component 2?Even along this component PGP can be differentiated from the other fining agents on the basis of terpenes, norisoprenoids and lactones
Line 294: it could be useful to indicate the kind of bentonite (sodium or calcium, natural or activated, etc)
Line 334: delete and
In conclusion, in my opinion, the present paper is suitable for needing only minor revisions.
Author Response
1. Just one general consideration let me uncertain about the possible commercial use of the PGP as fining agent that is the very high dosage needed to have an effect. I’m sure this issue has been already faced in the previous papers published on this topic, however for honesty this aspect should be at least mentioned also in the present paper. For example, using 13 mg/mL what is the loss of wine volume due to the adsorption by the fining agent? This aspect should be mentioned in the conclusions, as a possible limit for the applicability of this new product.
As the reviser has suggested, the authors have mentioned the commercial use of PGP and its implications in the conclusion. So the authors have added the following explanation “However, its use would need to be optimized since a loss of significant amount of wine volume through lees occurs following PGP addition. An option that will be tested is the repassing of wine over a fix bed of PGP to avoid wine losses.
Some additional suggestions:
2. Line 16: the sentence “ to further our knowledge in this respect” could be deleted. This sentence has been deleted in the paper yet.
3. Line 30: substitute and/or with “before or after”. The word and/or has been substituted by “before or after” in the paper.
4. Line 33: are you sure of the correctness of the citation number 5? Effectively, the citation number 5 was incorrected, so the authors have substituted by the correct citation.
5. Line 37: maybe is better “a reduction” instead of “the reduction”. As the reviser has suggested the authors have substituted “the reduction” by “ a reduction” in the paper.
6. Line 49 and 89: add the reference 23. The reference 23 has been added in the line 49 and 89.
7. Line 99, 117, 290, 335 finning à fining. In these lines have been substituted “finning” by “fining”.
8. Line 161: we observed. This part has been changed in the document.
9. Line 179: concentration of the last three…In the text has been substituted “concentration last three..” by “concentration of the last three”.
10. Line 190 (also in line 462): Vincenzi. It has been added the letter “n” to Vincenzi in lines 190 and 462.
11. Line 212: the. It has been corrected the mistake
12. Line 247: delete the last sentence (is a repetition).The sentence has been deleted in the document as the reviser has suggested.
13. Line 265-268: these data confirm those of Vincenzi who found an interaction of bentonite with fatty acids that increase with the length of the chain
This paragraph of the text has been changed.
14. Line 273: all the replicates of control wine together with those treated with… The text “all the control wines together with …” has been substituted by “all the replicates of the control wine together with those treated with…”
15. Line 274: maybe higher values (if it is the case, substitute also in line 275). What about the component 2? Even along this component PGP can be differentiated from the other fining agents on the basis of terpenes, norisoprenoids and lactones.
This paragraph has been rewritten following the suggestions made by the reviser.
16. Line 294: it could be useful to indicate the kind of bentonite (sodium or calcium, natural or activated, etc). The kind of bentonite used has been indicated in the document as the reviser has suggested.
17. Line 334: delete and. In the text has been deleted the word “and”.